# MultiCFV: Detecting Control Flow Vulnerabilities in Smart Contracts Leveraging Multimodal Deep Learning

## Abstract

The introduction of smart contract functionality marks the advent of the blockchain 2.0 era, enabling blockchain technology to support digital currency transactions and complex distributed applications. However, many smart contracts have been found to contain vulnerabilities and errors, leading to the loss of assets within the blockchain. Despite a range of tools that have been developed to identify vulnerabilities in smart contracts at the source code or bytecode level, most rely on a single modality, reducing performance, accuracy, and limited generalization capabilities. This paper proposes a multimodal deep learning approach, MultiCFV, which is designed specifically to analyze and detect erroneous control flow vulnerability, as well as identify code clones in smart contracts. Bytecode is generated from source code to construct control flow graphs, with graph embedding techniques extracting graph features. Abstract syntax trees are used to obtain syntax features, while code comments capture key commentary words and comment features. These three feature vectors are fused to create a database for code inspection, which is used to detect similar code and identify contract vulnerabilities. Experimental results demonstrate our method effectively combines structural, syntactic, and semantic information, improving the accuracy of smart contract vulnerability detection and clone detection.

## 1 Introduction

The concept of smart contracts was first introduced by computer scientist Nick Szabo in 1994 and gradually received significant attention with the emergence of Bitcoin (Nakamoto, 2008). A smart contract is an automated agreement that operates on blockchain technology, removing the need for third-party involvement. These contracts commonly involve transactions such as the transfer of cryptocurrency or digital assets, which are automatically executed when predefined conditions are met. This automation spans various fields, making smart contracts tamper-resistant and ensuring transparency and reliability in transactions (Kuo & Pham, 2023; Subramanian & Subramanian, 2022; Qi et al., 2023). However, the substantial value of the assets involved makes smart contracts prime targets for attackers looking to exploit vulnerabilities or errors in the contract's code. For instance, on October 7, 2023, the cryptocurrency exchange Mixin Network was hacked, resulting in a loss of approximately $200 million (Toulas, 2024).

Due to the immutable nature of smart contracts, they cannot be altered once deployed on the blockchain. Therefore, it is crucial to minimize vulnerabilities and errors in the code before deployment to enhance the security of the contract. Significant research efforts have led to advancements in blockchain systems and the development of tools designed to analyze and prevent smart contract vulnerabilities (He et al., 2023; di Angelo et al., 2023). Nevertheless, these tools still face several limitations. For instance, some tools require experts to define error patterns and detection rules that are not only time-consuming and labor-intensive but also struggle to address new or variant vulnerabilities effectively (Liu et al., 2021; Lin et al., 2023). To overcome the time-consuming and labor-intensive, certain tools utilize deep learning models to identify specific patterns or features associated with vulnerabilities (Wu et al., 2021; Yu et al., 2021; Gao, 2020). However, these tools primarily operate from the unimodal perspective, which often results in extracted features failing

to fully capture the semantic information, leading to reduced detection accuracy and compromised reliability (Adami, 2016).

In this paper, a novel approach MultiCFV is proposed to overcome these challenges through multimodal deep learning for smart contract clone detection and vulnerability verification. Our approach focuses on three key aspects: (1) Deep Learning Techniques: Deep learning is utilized to learn patterns and features of smart contract vulnerabilities, eliminating reliance on expert-defined detection rules and code design, which enables faster and more efficient detection. (2) Accuracy and Generalization: Detection accuracy and generalization capabilities are significantly enhanced through the use of multimodal deep learning. (3) Comment Information: To further improve detection accuracy, additional code information and features are extracted from comments within the code. These comments often provide insights into the function's purpose and considerations, offering valuable supplementary data.

The main contributions of this paper are as follows:

1. We propose a novel type of vulnerability and conduct an in-depth analysis. To the best of our knowledge, it's the first application of multimodal deep learning to smart contracts with erroneous control flow vulnerabilities.

2. We propose an innovative feature extraction approach by using multimodal deep learning and graph embedding techniques. Our approach overcomes the limitations of unimodal methods while enhancing detection accuracy and robustness. We integrate control flow graphs generated from bytecode, abstract syntax trees(AST) derived from source code, and code comments.

3. We introduce comment word embeddings as supplementary features for smart contracts. We highlight the importance of comments and include them in the feature set, thereby improving detection accuracy and overall performance.

4. We have uploaded the source code, experimental data, and comprehensive README documentation of MultiCFV. These resources ensure the reproducibility of our work and will be made open-source following the paper's publication.

Specifically, in the phase of code clone detection, the source code is not used directly as feature vectors. Instead, emphasis is placed on the control flow and semantic structure of smart contracts. Our approach avoids interference from irrelevant items such as variable and function names, leading to superior performance in both accuracy and generalization capability.

## 2 BACKGROUND

### 2.1 ERRONEOUS CONTROL FLOW VULNERABILITIES

Erroneous control flow vulnerabilities in smart contracts refer to design or implementation flaws that occur when handling exceptions or errors. These flaws can result in the smart contract failing to properly manage error situations, leading to unexpected behaviors or security issues.

Among the most common and severe erroneous control flow vulnerabilities in smart contracts is the reentrancy vulnerability (Xue et al., 2020; Wu et al., 2021). Reentrancy allows an attacker to repeatedly call a function during its execution, preventing the contract's state from being updated promptly and creating significant security risks. To prevent such vulnerabilities, smart contracts must rigorously verify the correctness and security of their behavior flows. In addition to reentrancy, other critical vulnerabilities include impermissible access control flaws, dangerous delegatecall vulnerabilities, and unchecked external call vulnerabilities (Zheng et al., 2024). Our detection focuses primarily on these four types of vulnerabilities. The rationale for focusing on these vulnerabilities is detailed in Appendix A.3.

### 2.2 CONTROL FLOW GRAPH

The Control Flow Graph (CFG) consists of basic blocks and control flow edges. Basic blocks are sequences of consecutive instructions in a program that contain no branches, representing a single execution unit (Contro et al., 2021). In this paper, basic blocks are composed of bytecode blocks formed by Ethereum Virtual Machine (EVM) instruction sequences. Branch instructions (e.g., JUMP,

JUMPI, RETURN) are the end markers of basic blocks, which are used to segment the basic blocks. Appendix A.5 provides a list of unique bytecode values along with their corresponding definitions and instructions.

Control flow edges refer to the transitions from one bytecode block to another based on conditional or call statements. In the CFG, different colors of control flow edges indicate different types of edges, mainly four types. An unconditional jump from one bytecode block to another is represented by a blue edge (unconditional edge, such as the JUMP instruction); the jump path when a conditional statement (conditional edge, such as JUMPI) is true is represented by a green edge; the jump path when a conditional statement is false is represented by a red edge (conditional edge, such as the JUMPI instruction); and the jump path involving calls to external functions within a bytecode block is represented by a yellow edge (function call edge).

## 3 METHODOLOGY

### 3.1 METHOD OVERVIEW

Erroneous control flow vulnerability is associated with smart contracts' behavioral logic and state transitions. Addressing this vulnerability requires a deep understanding of the contract's control flow and behavior across various states. However, a single graph alone cannot provide sufficient information. To obtain more adequate information, we apply multimodal deep learning to capture different features of smart contracts from three aspects: CFG, AST, and code comments.

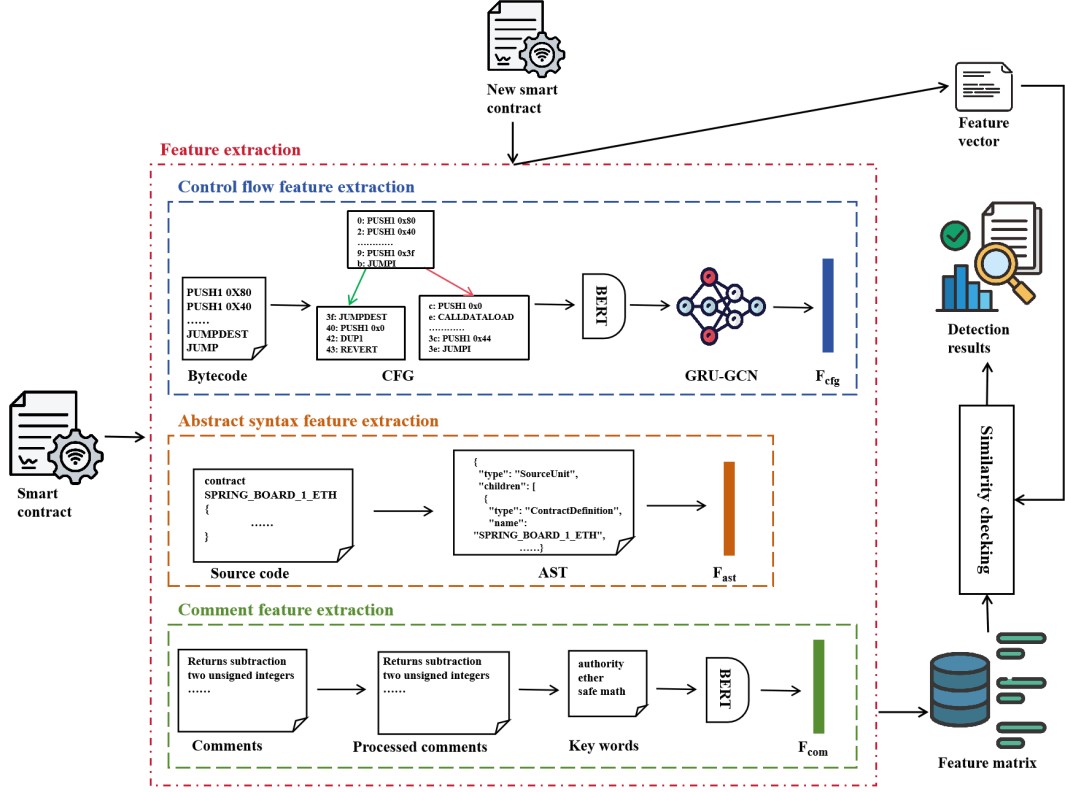

Figure 1: A High-level Overview of MultiCFV

The CFG, supplemented by the AST, provides valuable semantic and structural context. Code comments further offer insights into the contract's functionality and considerations, collectively enabling more effective features needed to identify erroneous control flow vulnerabilities. Our research considers the following aspects: (1) The CFG illustrates the control flow paths within the contract. By analyzing the graph, potential issues such as call errors, unchecked calls, and conditional

logic errors can be detected. This analysis helps in identifying control flow paths between basic blocks in the contracts, including conditional branches and jump paths. (2) The AST focuses on the structure and syntax of the code, including variable declarations, function definitions, and scopes. As a supplement to syntax and semantic checks, the AST facilitates error detection. (3) Combining CFG and AST enables a comprehensive analysis of smart contracts, leading to accurate detection and prevention of potential vulnerabilities in behavioral logic and state transitions.

Figure 1 presents the high-level overview of our approach MultiCFV, which comprises four key parts: control flow feature extraction, abstract syntax feature extraction, comment feature extraction, clone detection and contract verification. Specifically, bytecode is first generated from the source code, followed by the construction of the CFG by using this bytecode. A Graph Convolutional Network (GCN) combined with a Gated Recurrent Unit (GRU) is then employed to extract graph feature vectors from the CFG. The AST is extracted from the source code to obtain AST feature vectors. Additionally, key comment words and comment feature vectors are captured by utilizing attention mechanisms and fine-tuned BERT embeddings. Then, in the clone detection phase, these three feature vectors are integrated into a contract feature database for comparison with new input contracts. Contracts are considered as having similar codes if the similarity exceeds a defined threshold. In the contract verification phase, similarity measures are also used to assess new input contracts and detect erroneous control flow vulnerabilities.

## 3.2 CONTROL FLOW FEATURE EXTRACTION

### 3.2.1 EXTRACT CONTROL FLOW INFORMATION

The source code of contracts is converted into bytecode using a public compiler, and an automated tool called "Graphextractor" is developed to extract the CFG from the compiled bytecode. The extraction process is illustrated in Figure 2. Inspired by (Qian et al., 2023), the fine-tuned BERT model is used to process EVM instructions in the CFG, specifically named blocks, to extract features for these blocks as CFG node features.

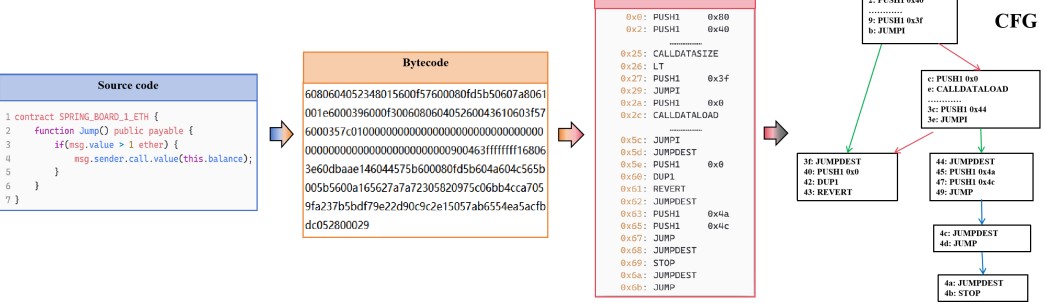

Figure 2: Control Flow Graph Extraction Process

The representation of control flow information for each contract is as follows.

$$O_{cfg} = (GP, NF, CN) \tag{1}$$

where $G$ contains all bytecode blocks and their corresponding control flow edges. $NF$ is the set of features for all bytecode blocks, represented by 256-dimensional vectors obtained from BERT embeddings, with each vector corresponding to a feature of a bytecode block. $CN$ is the name of the contract file, ending with '.sol'. The formula for $GP$ is as Equation 2.

$$GP = \{(u, g, v) \mid u, v \in V, g \in G\} \tag{2}$$

where $V$ is the set of nodes, and $G$ is the set of edge types.

The formula for node features NF is as follows.

$$NF = \begin{pmatrix} \mathbf{f}_1 & \mathbf{f}_2 & \mathbf{f}_3 & \cdots & \mathbf{f}_N \end{pmatrix}^{\top} \tag{3}$$

where $N$ is the total number of nodes, and each node's feature vector has a length of 128. $\mathbf{f}_i \in \mathbb{R}^{128}$ represents the feature vector of node $i$, i.e., $\mathbf{f}_i = (f_{i1}, f_{i2}, \ldots, f_{i128})$.

### 3.2.2 BERT EMBEDDING

The BERT model is employed as an embedding tool due to the contextual dependency of terms in EVM instructions and comments within smart contracts. BERT's contextual awareness can more accurately capture these dependencies (Jie et al., 2023). Moreover, smart contract code often involves complex logic and structure, and BERT's Transformer architecture is well-suited to capture and represent these intricate semantic details.

During feature extraction, the BERT model is not applied to all types of vulnerability contracts simultaneously. Instead, it is fine-tuned separately for each vulnerability type before being used for BERT embedding (Mosbach et al., 2020). This targeted approach optimizes the model for each vulnerability, improving the precision of feature extraction.

### 3.2.3 GRAPH EMBEDDING ON CFG

The graph embedding technique GRU-GCN is used to process control flow information and generate graph features from the CFG. The detailed rationale for selecting the GRU-GCN model, along with the complete process and formulas for generating the control flow graph features $F_{cfg} \in \mathbb{R}^{512}$ through graph embedding, are provided in Appendix A.4.

## 3.3 ABSTRACT SYNTAX FEATURE EXTRACTION

The process of extracting abstract syntax information is illustrated in Appendix A.2. An automated tool called "ast-generation" is developed to generate ASTs and extract key information from them. The extracted information is processed by a simple deep learning model to generate an abstract syntax feature vector for each contract, denoted as $F_{ast} \in \mathbb{R}^{512}$.

Appendix A.7 offers a detailed list of extracted roles and categories with their definitions. We primarily pay attention to the following aspects: the role of nodes, the role of their children, the number of child nodes, the presence of variables, the presence of input, and output parameters, etc.

## 3.4 COMMENT FEATURE EXTRACTION

The comment feature extractor operates as follows: comments are first extracted from the contract and cleaned to remove invalid characters, symbols, and meaningless words, retaining only relevant content. A convolutional neural network with a self-attention mechanism called " com-extractor " extracts keywords and feature vectors from the comments. The number of keywords depends on the comment length and represents the contract. The comment feature extraction process is the same as detailed in Section 3.2.2, producing a comment feature vector for each contract, denoted as $F_{ast} \in \mathbb{R}^{512}$. The specific rationale for selecting a convolutional neural network with a self-attention mechanism is detailed in Appendix A.8.

What's more, we compile the keywords from all comments and generate a word cloud, as shown in Figure 3. The figure shows that most of these contracts use the SafeMath library, and a large portion of the code content involves mathematical operations (Hefele et al., 2019).

## 3.5 CONTRACT VERIFICATION AND CLONE DETECTION

The CFG feature vector $F_{cfg}$, the AST feature vector $F_{ast}$, and the comment feature vector $F_{com}$ described above are vertically stacked to form the comprehensive feature representation matrix $\mathbf{F}$ for each smart contract, which is used to verify the presence of erroneous control flow vulnerabilities and similar code. The comprehensive feature representation matrices $\mathbf{F}$ for all smart contracts are stored in a database for code clone detection. Due to the high-dimensional and sparse nature of the data, the RBF kernel function is multiplied with cosine similarity for code similarity computation. The detailed reasons are listed in Appendix A.9.

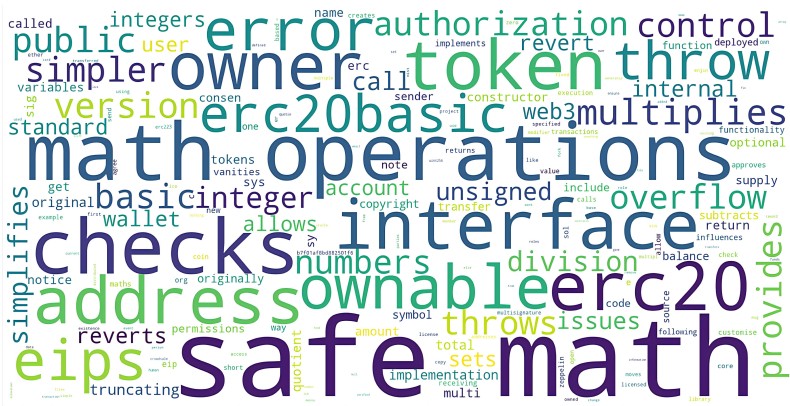

Figure 3: Comment Wordcloud

# 4 EXPERIMENTS

## 4.1 DATA COLLECTION

To obtain a large number of smart contracts, we select four distinct datasets: Smartbugs Curated (di Angelo et al., 2023), SolidiFI-Benchmark (Ghaleb & Pattabiraman, 2020), MessiQ-Dataset (Qian et al., 2023; Liu et al., 2023), and Clean Smart Contracts from Smartbugs Wild (Nguyen et al., 2022). Detailed information on these four datasets is provided in Appendix A.6.

## 4.2 EXPERIMENTAL SETUP

The GRU-GCN, and "com-extractor" are implemented using PyTorch. GRU-GCN has a hidden layer size of 512 and consists of a convolutional layer, two dropout layers, three GRU layers, a fully connected layer, and a regression layer. The learning rate is 0.0001, and the Adam optimizer is used for training. Moreover, the "com-extractor" has a hidden layer size of 512 with 4 convolutional layers.

The dataset is split with an 8:2 ratio for contract vulnerability and code clone detection. Due to the dataset's imbalance from an abundance of negative samples in vulnerability detection, a balanced dataset is created using SMOTE and data augmentation. The model processes three modalities: comment features, AST features, and CFG features, employing Binary Cross-Entropy Loss and the Adam optimizer with a 0.005 learning rate. To reduce overfitting, Dropout regularization (probability 0.3) is applied. Features from all modalities are concatenated and passed through a fully connected layer with ReLU activation, followed by a sigmoid layer to output probabilities. During 500 epochs, the model with the lowest loss is saved for evaluation. Vulnerabilities are identified if probabilities exceed 0.95.

## 4.3 ABLATION EXPERIMENTS

Given the wide range of vulnerabilities detected in this study, Reentrancy vulnerability, one of the most common types, is selected as the reference for ablation experiments.

### 4.3.1 LEARNING RATE SELECTION

A comparative analysis of different learning rates is conducted to determine the optimal value for achieving the best performance. Table 1 shows that the model achieves the highest accuracy and performance at a learning rate of 0.005. Meanwhile, the ROC curve for our approach's detection results is illustrated in Figure 4, showing an AUC of 0.9947.

### 4.3.2 MULTI-MODAL INTEGRATION

Ablation studies highlight the essential role of multimodal deep learning in the proposed approach. Models trained on single features (e.g., comment, AST, or CFG) or dual-modal combinations

| Learning Rate | ACC | RE | PRE | F1 |
|---|---|---|---|---|
| 0.01 | 98.25 | 99.42 | 97.15 | 98.27 |
| 0.005 | **99.13** | **98.25** | **98.65** | **98.45** |
| 0.001 | 98.35 | 99.03 | 97.70 | 98.36 |
| 0.0005 | 98.16 | 99.42 | 96.97 | 98.18 |

Table 1: Performance Comparison (%) Across Different Learning Rates in Terms of Accuracy (ACC), Recall (RE), Precision (PRE), and F1-Score (F1)

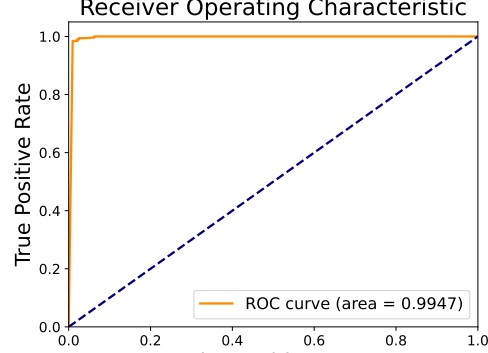

Figure 4: ROC Curve of MultiCFV Detection Results

| Modality | ACC | RE | PRE | F1 |
|---|---|---|---|---|
| Comments | 52.04 | 36.12 | 75.00 | 48.75 |
| AST | 62.33 | 50.68 | 89.38 | 64.68 |
| CFG | **85.74** | **92.49** | **91.77** | **92.13** |
| AST & CFG | 96.88 | 95.48 | 92.99 | 94.22 |
| AST & Comments | 71.84 | 54.05 | 97.07 | 61.54 |
| CFG & Comments | **90.45** | **94.26** | **92.75** | **93.45** |
| All | **99.13** | **98.25** | **98.65** | **98.45** |

Table 2: Performance Comparison (%) between Single-Modal (AST, CFG, or Comments Features), Dual-Modal (AST + CFG, AST + Comments, or CFG + Comments), and Multi-Modal (AST + CFG + Comments) Approaches in Vulnerability Detection in Terms of ACC, RE, PRE, and F1

(e.g., AST & CFG or comment & AST) consistently underperformed the multimodal approach, underscoring the complementary benefits of integrating multiple modalities. Experimental results show that CFG achieved the highest accuracy among single-modal features, while CFG & AST outperformed other dual-modal pairings. However, neither single-modal nor dual-modal setups matched the performance of the fully integrated multimodal approach.

### 4.4 CONTRACT VERIFICATION

According to Zheng et al., Slither and Mythril currently exhibit the highest accuracy in contract vulnerability detection (Zheng et al., 2024; Josselin, 2024; Bast, 2024). Comparative experiments are conducted using the Slither and Mythril tool. Table 3 presents comparative experiments using these tools. Notably, Slither failed to analyze 66 contracts, while Mythril encountered even more failures, primarily due to limitations related to the supported ranges of Solidity compiler versions.

Additionally, MultiCFV is tested on a new vulnerability dataset (Unprotected Ether Withdrawal). The detection results showed an accuracy of 82.86%, a precision of 92.07%, a recall of 83.34%, and an F1-score of 87.49%. This demonstrates that MultiCFV is very generalizable.

| Tool | Reentrancy | | | | Access Control | | | | External Call | | | | Delegatecall | | | |
|------|-----|-----|-----|-----|-----|-----|-----|-----|-----|-----|-----|-----|-----|-----|-----|-----|
| | ACC | RE | PRE | F1 | ACC | RE | PRE | F1 | ACC | RE | PRE | F1 | ACC | RE | PRE | F1 |
| Mythril | 66.17 | 64.33 | 65.09 | 64.71 | 0 | 0 | 0 | 0 | 50.14 | 51.25 | 54.01 | 52.59 | 59.56 | 60.59 | 61.40 | 60.99 |
| Slither | 72.76 | 71.11 | 73.67 | 72.36 | 0 | 0 | 0 | 0 | 63.12 | 60.56 | 66.20 | 63.22 | 66.91 | 67.97 | 69.44 | 68.70 |
| Slither & Mythril | 76.71 | 77.37 | 77.03 | 77.20 | 0 | 0 | 0 | 0 | 64.39 | 63.84 | 67.67 | 65.71 | 68.46 | 68.52 | 70.26 | 69.38 |
| MultiCFV | **99.13** | **98.25** | **98.65** | **99.12** | **82.89** | **92.39** | **89.77** | **91.06** | **89.01** | **98.17** | **90.16** | **93.99** | **80.71** | **90.06** | **81.61** | **85.63** |

Table 3: Performance Comparison (%) between MultiCFV and Slither in Terms of ACC, RE, PRE, and F1

### 4.5 CODE CLONE DETECTION

In code clone detection, MultiCFV identifies contracts with similarity scores above a specified threshold and outputs their names and contents. A randomly selected smart contract is used for this analysis, with detailed contract content and detection results provided in Appendix A.10. Additionally, we compare the performance of SmartEmbed and MultiCFV on the same dataset (Gao et al., 2020) to evaluate the effectiveness of MultiCFV. Detection times are averaged over five runs for each threshold value, with results in Table 4 showing that MultiCFV slightly outperforms SmartEmbed in terms of speed. What's more, Venn diagrams of the experimental results are plotted at a similarity threshold of 0.95, as illustrated in Figure 5. SE represents SmartEmbed, MT represents MultiCFV, MT_rest indicates the similar codes detected by MultiCFV but not by SmartEmbed, and SE_rest indicates the similar codes detected by SmartEmbed but not by MultiCFV. SmartEmbed failed to detect clone codes in 26% of the contract codes, whereas MultiCFV only in 17%. Along with Figure 5, it is indicated that SmartEmbed is overly cautious in clone detection, potentially overlooking codes with similar structures and functions. In contrast, our approach imposes fewer constraints on the syntax and compilation versions of the contracts, resulting in more effective detection. We also plot a Venn diagram with a similarity threshold of 1.0, which is presented in Appendix A.11.

| Threshold | Tool | Average Time(s) |
|-----------|------|-----------------|
| 0.95 | SmartEmbed | 403.7637 |
| | MultiCFV | **368.6572** |
| 1 | SmartEmbed | 396.9978 |
| | MultiCFV | **356.8932** |

Table 4: The Detection Time of Code Clone

It is important to note that detecting clone codes with identical structures and functions does not always equate to better performance when there is a higher overlap. In the remaining dataset, variations in variable names, function names, and other code elements introduce differences, as illustrated in Figure 8 in Appendix A.10.

## 5 RELATED WORK

### 5.1 VULNERABILITY DETECTION

Deep learning has significantly advanced vulnerability detection in smart contracts, enhancing performance. Yu et al. introduced Deescvhunter, a deep learning framework for automatic vulnerability detection (Yu et al., 2021). Liu et al. combined expert knowledge with graph neural networks to improve contract vulnerability detection (Liu et al., 2021). Wu et al. developed Peculiar, which detects reentrancy vulnerabilities using control flow graphs and graph neural networks (Wu et al., 2021). Similarly, Chen et al. and Zhuang et al. employed control flow graphs and graph neural networks for detecting diverse vulnerabilities (Chen et al., 2024; Zhuang et al., 2021). Cai et al. further integrated control flow graphs, abstract syntax trees, and program dependency graphs, leveraging graph neural networks for feature extraction (Cai et al., 2023). These methods highlight the effectiveness of graph embedding in preserving structural information and enhancing detection accuracy.

Despite these advancements, the limitations of unimodal methods have driven the adoption of multimodal approaches. Jie et al. proposed a multimodal framework for detecting contract vulnerabilities (Jie et al., 2023), while Qian et al. introduced a cross-modality mutual learning framework,

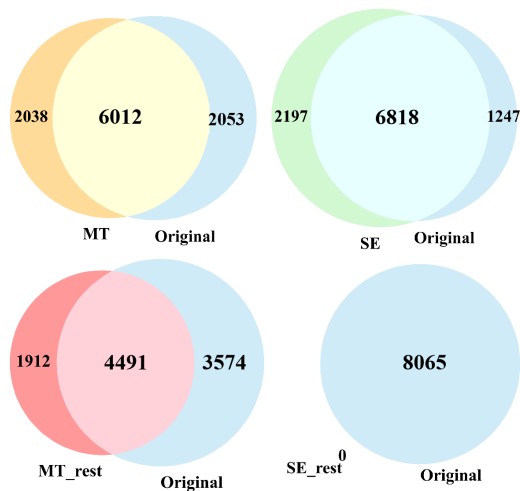

Figure 5: Venn Diagram for Clones Detected by MultiCFV and SmartEmbed with Similarity Threshold 0.95

showing that multimodal methods outperform unimodal ones (Qian et al., 2023). Wang et al. developed SMARTINV, a cross-modal tool for identifying vulnerabilities by checking invariant violations (Wang et al., 2024b). These approaches integrate information from multiple modalities, achieving a more comprehensive understanding of vulnerabilities (Yang et al., 2021).

However, these methods rely on multi-class classification tasks and cannot achieve multiple downstream tasks like clone detection. They also lack generalization capabilities and struggle to adapt to new vulnerability patterns.

## 5.2 CODE CLONE DETECTION

Kondo et al. reported that 79.2% of smart contracts are clones, with the number of clones rapidly increasing (Kondo et al., 2020). Similarly, He et al.(He et al., 2020) and Chen et al.(Chen et al., 2021) observed high code reuse rates, highlighting the critical need for clone detection to ensure smart contract security and enable thorough analysis. To address this, Kondo et al.(Kondo et al., 2020) developed Deckard, a tree-based clone detection tool, while Gao et al.(Gao et al., 2020) introduced SmartEmbed, a Word2vec-based tool that outperformed Deckard. Further advancements include Wang et al.'s (Wang et al., 2024a) SolaSim, leveraging weighted control flow graphs, and Ashizawa et al.'s (Ashizawa et al., 2021) Eth2Vec, designed for code-rewriting clone detection.

However, these methods share a common issue: they fail to effectively preserve the structural information and features of the code. The extracted features do not fully represent the contract's structure and variable scope. Moreover, some methods, such as SmartEmbed and Deckard, use partial technologies, resulting in suboptimal performance in retaining code semantics and structure.

## 6 CONCLUSION AND FUTURE PERSPECTIVES

We propose MultiCFV, a multimodal deep learning-based approach for contract verification and code clone detection, achieving superior generalization and accuracy. As the first to apply multimodal deep learning to this domain, MultiCFV identifies erroneous control flow vulnerabilities and detects code similarities between new input code and existing code, highlighting similar segments. It outperforms Slither and Mythril across metrics such as accuracy, precision, and F1-score, while effectively identifying similar contracts in clone detection. However, MultiCFV is currently limited to contract-level clone detection, which is relatively coarse-grained. Future work will aim to develop finer-grained detection methods to improve precision and practical applicability.

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

# A APPENDIX

## A.1 THE USE OF LARGE LANGUAGE MODELS (LLMS)

During the paper writing process, LLMs have been utilized for English translation and polishing.

## A.2 AST EXTRACTION PROCESS

The extraction process of AST is illustrated in Figure 6.

## A.3 RATIONALE FOR FOCUSING ON THE FOUR SPECIFIC VULNERABILITIES

We selected these four vulnerabilities for the following reasons: (i) In real-world attacks, 70% of financial losses in Ethereum smart contracts are caused by these vulnerabilities (Chen et al., 2020). (ii) Existing research indicates that these vulnerabilities are more prevalent in Ethereum smart contracts, especially in newer versions of smart contract code. Studies have shown that contracts

Figure 6: AST Information Extraction Process

compiled with post-2020 compiler versions (i.e., versions higher than 0.6) are particularly susceptible to these vulnerabilities (Gao et al., 2019; Praitheeshan et al., 2019; Rodler et al., 2018). Zheng et al. (Zheng et al., 2024) found that more than 50% of the code containing these four types of vulnerabilities was present in 66.5% of high-version contract compilers. (iii) These vulnerabilities represent typical erroneous control flow issues. For instance, a lack of permission control leads to erroneous control flow (as seen in delegatecall and impermissible access control flaws vulnerabilities), insufficient attention to inter-contract interactions results in erroneous control flow (as in reentrancy vulnerabilities), and unchecked or inadequately checked external calls lead to erroneous control flow (as in unchecked external call vulnerabilities).

## A.4 RATIONALE FOR GRU-GCN

The detailed reason for choosing GRU-GCN is based on the following considerations:(1) Effective Capture of Local Structural Information: GCN updates the representation of each node by aggregating information from neighboring nodes, effectively capturing local structural information and features of

the nodes. It encodes the topological relationships in the graph as vector representations, preserving the structural characteristics of the graph in the vector space (Zhuang et al., 2021; Liu et al., 2021). This representation is particularly suited for downstream tasks such as code similarity analysis and vulnerability detection, aligning well with our research objectives. (2) Dynamic Adjustment of Feature Weights: When processing the node features generated by GCN, GRU can dynamically adjust the weights of the features and retain important sequential information. This allows the model to focus on nodes and edges more relevant to the current task, enhancing its ability to capture complex relationships between nodes, improving learning effectiveness, and mitigating the risk of overfitting.

### A.4.1 ALGORITHEM OF OBTAINING THE OUTPUT CONTROL FLOW GRAPH FEATURE VECTOR

The calculation of the node feature matrix $\mathbf{H}^{(1)}$ output from the graph convolution layer is as follows:

$$\mathbf{H}^{(1)} = \text{ReLU}(\hat{A}\mathbf{O_{cfg}}\mathbf{W}^{(1)}) \tag{4}$$

Here, $\mathbf{H}^{(1)}$ has the shape $N_{\text{batch}} \times D_{\text{output}}$, where $N_{\text{batch}}$ is the batch size, representing the number of contracts in the batch, set to 1024. $D_{\text{output}}$ is the dimension of the output features, set to 512. $\mathbf{O_{cfg}}$ is the input control flow feature matrix of the contract with the shape $N_{\text{batch}} \times D_{\text{input}}$, where $D_{\text{input}}$ is the dimension of the input features. $\hat{A}$ is the normalized adjacency matrix, and $\mathbf{W}^{(1)}$ is the weight matrix for the graph convolution layer. ReLU denotes the rectified linear unit activation function.

The GRU computes the hidden state for each node. The update gate determines the proportion of the current hidden state combined with the previous hidden state and the new candidate hidden state:

$$\mathbf{z}_t = \sigma(\mathbf{W}_z\mathbf{x}_t + \mathbf{U}_z\mathbf{h}_{t-1}) \tag{5}$$

where $\sigma$ is the nonlinear activation function, $\mathbf{W}_z$ is the weight matrix for the update gate input, and $\mathbf{U}_z$ is the weight matrix from the previous time step's hidden state to the update gate. $\mathbf{x}_t$ is the input at the current time step $t$, and $\mathbf{h}_{t-1}$ represents the hidden state at the previous time step $t-1$.

The reset gate determines the extent to which the previous hidden state influences the calculation of the new candidate's hidden state:

$$\mathbf{r}_t = \sigma(\mathbf{W}_r\mathbf{x}_t + \mathbf{U}_r\mathbf{h}_{t-1}) \tag{6}$$

where $\mathbf{W}_r$ is the weight matrix for the reset gate input, and $\mathbf{U}_r$ is the weight matrix from the previous time step's hidden state to the reset gate.

The new candidate hidden state is computed as follows, incorporating the reset gate's output to reflect the combined information of the current input and the previous hidden state:

$$\tilde{\mathbf{h}}_t = \tanh(\mathbf{W}\mathbf{x}_t + \mathbf{r}_t \odot \mathbf{U}\mathbf{h}_{t-1}) \tag{7}$$

where tanh is the hyperbolic tangent activation function, $\mathbf{W}$ is the weight matrix for the new candidate hidden state, and $\mathbf{U}$ is the weight matrix from the previous hidden state to the new candidate hidden state. $\odot$ denotes element-wise multiplication.

The final hidden state is computed as follows:

$$\mathbf{h}_t = (1 - \mathbf{z}_t) \odot \mathbf{h}_{t-1} + \mathbf{z}_t \odot \tilde{\mathbf{h}}_t \tag{8}$$

Here, $\mathbf{H}^{(2)}$ has the shape $N_{\text{batch}} \times D_{\text{hidden}}$, with $D_{\text{hidden}}$ set to 512.

$$\mathbf{H}^{(2)} = \begin{pmatrix} \mathbf{h}_1 & \mathbf{h}_2 & \mathbf{h}_3 & \cdots & \mathbf{h}_N \end{pmatrix}^{\top} \tag{9}$$

Finally, we input $\mathbf{H}^{(2)}$ into fully connected and regression layers to obtain the output control flow graph feature vector $F_{cfg} \in \mathbb{R}^{512}$.

### A.5 DEFINITIONS OF BYTECODE VALUES AND INSTRUCTIONS

The 11 categories of bytecode values and their corresponding definitions are presented, along with the distinctive opcodes used as features to represent the binary instruction operations.

### A.5.1 STOP AND ARITHMETIC OPERATIONS

- **0x00 - 0x0B**:
  - 0x00 - STOP
  - 0x01 - ADD
  - 0x02 - MUL
  - 0x03 - SUB
  - 0x04 - DIV
  - 0x05 - SDIV
  - 0x06 - MOD
  - 0x07 - SMOD
  - 0x08 - ADDMOD
  - 0x09 - MULMOD
  - 0x0A - EXP
  - 0x0B - SIGNEXTEND

### A.5.2 COMPARISON AND BITWISE LOGIC OPERATIONS

- **0x10 - 0x1A**:
  - 0x10 - LT
  - 0x11 - GT
  - 0x12 - SLT
  - 0x13 - SGT
  - 0x14 - EQ
  - 0x15 - ISZERO
  - 0x16 - AND
  - 0x17 - OR
  - 0x18 - XOR
  - 0x19 - NOT
  - 0x1A - BYTE
  - 0x1B - SHL
  - 0x1C - SHR
  - 0x1D - SAR

### A.5.3 KECCAK256 METHOD

- **0x20**:
  - 0x20 - KECCAK256

### A.5.4 ENVIRONMENTAL INFORMATION

- **0x30 - 0x3E**:
  - 0x30 - ADDRESS
  - 0x31 - BALANCE
  - 0x32 - ORIGIN
  - 0x33 - CALLER
  - 0x34 - CALLVALUE
  - 0x35 - CALLDATALOAD
  - 0x36 - CALLDATASIZE
  - 0x37 - CALLDATACOPY
  - 0x38 - CODESIZE
  - 0x39 - CODECOPY
  - 0x3A - GASPRICE

- 0x3B - EXTCODESIZE
- 0x3C - EXTCODECOPY
- 0x3D - RETURNDATASIZE
- 0x3E - RETURNDATACOPY

### A.5.5 BLOCK INFORMATION

- **0x40 - 0x45**:
  - 0x40 - BLOCKHASH
  - 0x41 - COINBASE
  - 0x42 - TIMESTAMP
  - 0x43 - NUMBER
  - 0x44 - DIFFICULTY
  - 0x45 - GASLIMIT
  - 0x46 - CHAINID

### A.5.6 STACK, MEMORY, STORAGE AND FLOW OPERATIONS

- **0x50 - 0x5B**:
  - 0x50 - POP
  - 0x51 - MLOAD
  - 0x52 - MSTORE
  - 0x53 - MSTORE8
  - 0x54 - SLOAD
  - 0x55 - SSTORE
  - 0x56 - JUMP
  - 0x57 - JUMPI
  - 0x58 - PC
  - 0x59 - MSIZE
  - 0x5A - GAS
  - 0x5B - JUMPDEST

### A.5.7 PUSH OPERATIONS

- **0x60 - 0x7F**:
  - 0x60 - PUSH1
  - 0x61 - PUSH2
  - ...
  - 0x7F - PUSH32

### A.5.8 DUPLICATION OPERATIONS

- **0x80 - 0x8F**:
  - 0x80 - DUP1
  - 0x81 - DUP2
  - ...
  - 0x8F - DUP16

### A.5.9 EXCHANGE OPERATIONS

- **0x90 - 0x9F**:
  - 0x90 - SWAP1
  - 0x91 - SWAP2
  - ...
  - 0x9F - SWAP16

### A.5.10 LOGGING OPERATIONS

- **0xA0 - 0xA4**:
  - 0xA0 - LOG0
  - 0xA1 - LOG1
  - 0xA2 - LOG2
  - 0xA3 - LOG3
  - 0xA4 - LOG4

### A.5.11 SYSTEM OPERATIONS

- **0xF0 - 0xFF**:
  - 0xF0 - CREATE
  - 0xF1 - CALL
  - 0xF2 - CALLCODE
  - 0xF3 - RETURN
  - 0xF4 - DELEGATECALL
  - 0xF5 - CREATE2
  - 0xFA - STATICCALL
  - 0xFD - REVERT
  - 0xFE - INVALID
  - 0xFF - SELFDESTRUCT

## A.6 DETAILED INFORMATION ABOUT THE DATASET

The detailed information on the four distinct datasets is as follows:

1. Smartbugs Curated (di Angelo et al., 2023): This dataset is one of the most commonly used real-world datasets for automatic reasoning and testing of Solidity smart contracts. It includes 143 annotated contracts with a total of 208 vulnerabilities.

2. SolidiFI-Benchmark (Ghaleb & Pattabiraman, 2020): This synthetic dataset contains vulnerable smart contracts. It comprises 350 different contracts with 9,369 injected vulnerabilities, covering seven different vulnerability types.

3. MessiQ-Dataset (Qian et al., 2023; Liu et al., 2023): This is the most recent dataset with the highest variety of vulnerabilities, containing 12,000 vulnerable smart contracts, which can be downloaded at `https://drive.google.com/file/d/1iU2J-BIstCa3ooVhXu-GljOBzWi9gVrG/view`

4. Clean Smart Contracts from Smartbugs Wild (Nguyen et al., 2022): Based on the results of 11 integrated detection tools, the Smartbugs framework identified 2,742 out of 47,398 contracts as free of errors. These 2,742 contracts are used as a set of clean contracts for comparison purposes.

## A.7 AST INFORMATION

Here are all the types we extracted, including contract types, function types, and variable types: "StateVariableDeclaration", "EmitStatement", "contract", "Conditional", "FunctionCall", "NumberLiteral", "ThrowStatement", "ExpressionStatement", "MemberAccess", "ReturnStatement", "IndexAccess", "ForStatement", "StringLiteral", "interface", "TupleExpression", "BooleanLiteral", "IfStatement", "ModifierDefinition", "StructDefinition", "EventDefinition", "InlineAssemblyStatement", "WhileStatement", "library", "Identifier", "UnaryOperation", "VariableDeclarationStatement", "PragmaDirective", "BinaryOperation", "ElementaryTypeNameExpression", "EnumDefinition", "ContractDefinition", "FunctionDefinition", "UsingForDeclaration", "block".

We categorized these types into different classes and explained the specific meaning of each type.

### A.7.1 CONTRACT STRUCTURE RELATED

- ContractDefinition - A contract definition node, representing a smart contract.

- contract - Indicates this is a regular contract.
- interface - Indicates this is an interface.
- library - Indicates this is a library.

- StructDefinition - A struct definition node, representing a structure.
- EnumDefinition - An enum definition node, representing an enumeration.
- StateVariableDeclaration - A state variable declaration node, representing a state variable.
- EventDefinition - An event definition node, representing an event.
- ModifierDefinition - A modifier definition node, representing a function modifier.
- UsingForDeclaration - A using statement node, representing a using for declaration.

### A.7.2 Function Related

- FunctionDefinition - A function definition node, representing a function.
- ReturnStatement - A return statement node, representing a 'return' statement.
- ThrowStatement - A throw statement node, representing a 'throw' statement.
- EmitStatement - An emit statement node, representing an 'emit' statement.
- FunctionCall - A function call node, representing a function call.

### A.7.3 Expression Related

- ExpressionStatement - An expression statement node, representing an expression.
- MemberAccess - A member access node, representing access to a member of an object (e.g., object.member).
- IndexAccess - An index access node, representing access to an array or mapping index (e.g., array[index]).
- TupleExpression - A tuple expression node, representing a tuple (e.g., (a, b)).
- UnaryOperation - A unary operation node, representing a unary operation (e.g., -a).
- BinaryOperation - A binary operation node, representing a binary operation (e.g., a + b).
- Conditional - A conditional expression node, representing a ternary operator (e.g., a ? b : c).
- ElementaryTypeNameExpression - An elementary type name expression node, representing a basic type (e.g., uint256).An elementary type name expression node, representing a basic type (e.g., uint256).

### A.7.4 Literal Related

- NumberLiteral - A number literal node, representing a number (e.g., 123).
- StringLiteral - A string literal node, representing a string (e.g., "hello").
- BooleanLiteral - A boolean literal node, representing a boolean value (e.g., true or false).

### A.7.5 Statement Related

- IfStatement - An if statement node, representing an 'if' statement
- ForStatement - A for statement node, representing a 'for' loop.
- WhileStatement - A while statement node, representing a 'while' loop.
- InlineAssemblyStatement - An inline assembly statement node, representing an inline assembly block.
- VariableDeclarationStatement - A variable declaration statement node, representing a variable declaration.

### A.7.6 Identifier Related

- Identifier - An identifier node, representing the name of a variable or object.

### A.7.7 OTHERS

- PragmaDirective - A pragma directive node, representing a pragma directive (e.g., 'pragma solidity ô.8.0').

- block - Represents a block of code.

## A.8 RATIONALE FOR UTILIZING SELF-ATTENTION MECHANISM WITH A CONVOLUTIONAL NEURAL NETWORK

based on the following considerations: (1) The self-attention mechanism can identify relationships between distant words in the comments, which may be important for keyword extraction (Alammary, 2022). (2) The self-attention mechanism can assign different weights to each word in the sequence, reflecting the importance of the words and more accurately identifying the keywords (Nozza et al., 2020). (3) Combining the self-attention mechanism with a convolutional neural network allows for the extraction of local features (through the convolutional layers) while enhancing the global semantic representation (through the self-attention mechanism), thereby providing a more comprehensive understanding of the text.

## A.9 RATIONALE FOR MULTIPLYING RBF KERNEL AND COSINE SIMILARITY

The detailed reasons for using the multiplication of the RBF kernel function and cosine similarity are as follows: (1) The RBF kernel function captures nonlinear relationships in the input data by computing similarities in a high-dimensional space, thus handling complex relationships and patterns more effectively. Additionally, the RBF kernel is robust to noise and variations in the input data. It calculates similarity by considering the distance between input data and the central point, which effectively manages minor variations and noise. (2) According to Yuan et al. (Yuan et al., 2023), cosine similarity is well-suited for high-dimensional sparse data, making it particularly appropriate for comparing texts or feature vectors. (3) The product computation ensures that if either similarity measure is low, the final result is also low. This guarantees that high similarity is achieved only when both measures are high, thereby enhancing the accuracy of similarity computation.

## A.10 EXAMPLE OF CODE CLONE DETECTION

Figure 7 shows the randomly input contract content used for clone detection. Figure 8 displays the content of two contracts from the clone detection output results.

```
1 contract ELTWagerLedger {
2     mapping (address => mapping (address => uint)) public tokens;
3     function withdraw(uint amount) {
4         if (tokens[0][msg.sender] < amount) throw;
5         if (!msg.sender.call.value(amount)()) throw;
6         tokens[0][msg.sender] = tokens[0][msg.sender] - amount;
7     }
8 }
```

Figure 7: Example Code of Code Clone Detection

```
1 contract ELTWagerLedger {
2     mapping (address => mapping (address => uint)) public tokens;
3     function withdraw(uint amount) {
4         if (tokens[0][msg.sender] < amount) throw;
5         if (!msg.sender.call.value(amount)()) throw;
6         tokens[0][msg.sender] = tokens[0][msg.sender] - amount;
7     }
8 }                              result1
```

```
1 contract Private_Bank {
2     mapping (address => uint) public balances;
3     function CashOut(uint _am) {
4         if(_am <= balances[msg.sender]) {
5             if(msg.sender.call.value(_am)()){
6                 balances[msg.sender] -= _am;
7             }
8         }
9     }
10 }                             result2
```

Figure 8: Two Contracts from the Code Clone Detection Results

## A.11 VENN DIAGRAM

Figure 9 is a Venn diagram with a similarity threshold of 1.0 between MultiCFV and SmartEmbed. SE represents SmartEmbed, MT represents MultiCFV, MT_rest indicates the similar codes detected by MultiCFV but not by SmartEmbed, and SE_rest indicates the similar codes detected by SmartEmbed but not by MultiCFV. SmartEmbed failed to detect clone codes in 26% of the contract codes, whereas MultiCFV only in 17%.

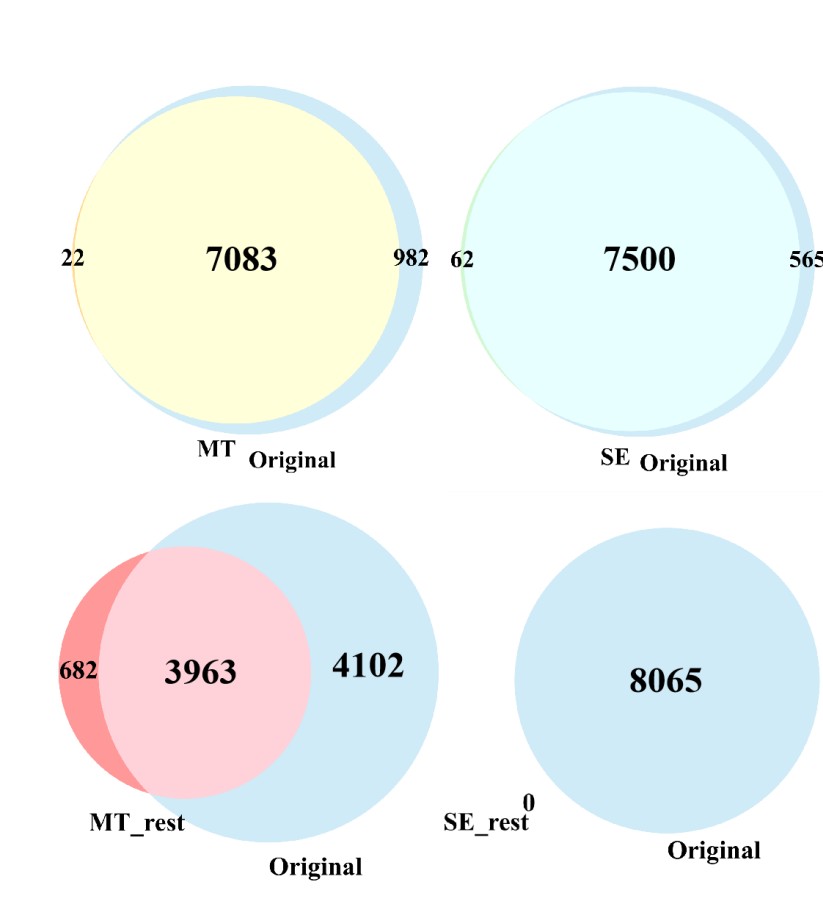

Figure 9: Venn Diagram for Clones Detected by MultiCFV and SmartEmbed with Similarity Threshold 1.0

