# OpenReview forum: "MultiCFV: Detecting Control Flow Vulnerabilities in Smart Contracts Leveraging Multimodal Deep Learning"
_ICLR.cc/2026/Conference — Submitted to ICLR 2026_

### Official Review · Reviewer_JHuL · 2025-10-22

**Soundness:** 2
**Presentation:** 2
**Contribution:** 2
**Rating:** 4
**Confidence:** 4

**Summary:**

The paper introduces MultiCFV, a deep learning framework for detecting control-flow-related vulnerabilities and code clones in smart contracts. It combines control-flow graphs (CFG) extracted from bytecode and abstract syntax trees (AST) from source code, and exploit a GRU-GCN and another independent network to process them. Moreover, comment embeddings encoded by fine-tuned BERT models are also used.The three input features are concatenated for final prediction. Experiments on four public datasets show the proposed model outperforming existing static tools.

**Strengths:**

- This work focuses on a pratical problem and address real-world vulnerabilities.
- The combination of structural (CFG), syntactic (AST), and semantic (comments) information in a framework is a contribution.
- Implementation details and source code are provided.

**Weaknesses:**

- The proposed architecture integress three components (BERT, GCN, and another network) to process three different features (comment, CFG, and AST). However, all these techniques have already been well explored and widely used in existing approaches. This work represents an incremental extension of earlier multi-encoder frameworks, rather than aligning with the current frontier of LLM-driven contract analysis.
- Some implementation details are missing. For example, the AST feature is extracted by a deep learning model but the architecture is not clearly provided.
- The paper does not include any baseline or discussion involving modern LLM-based approaches.
- The paper claims utilizing deep learning techniques can  enable faster and more efficient detection, but there are no measurements of inference time or computational cost on the vulnerability detection task.
- The writings need improving. There are several grammatical errors and typos (e.g., To overcome the time-consuming and labor-intensive,).

**Questions:**

- Please improve the writing quality.
- Could the method generalize to function-level or statement-level vulnerability detection instead of contract-level?
- Please clarify the computational cost of MultiCFV compared to the compared baselines on the vulnerability detection task.
- It would strengthen the paper to include a comparison with modern LLMs and explain why a finetuned GCN+BERT architecture remains necessary in the current LLM era.
- It would benifit the paper if evaluation on real-world deployed contracts is provided.

---

### Official Review · Reviewer_Neod · 2025-10-31

**Soundness:** 2
**Presentation:** 2
**Contribution:** 2
**Rating:** 4
**Confidence:** 3

**Summary:**

This paper proposes MultiCFV, a multimodal deep learning method for detecting control-flow-related vulnerabilities and code clones in smart contracts. The approach integrates three modalities: Control Flow Graphs (CFG) for structural features, Abstract Syntax Trees (AST) for syntactic features, and code comments for semantic information. The features from these modalities are fused to train a model for vulnerability detection and to build a feature database for clone detection.

**Strengths:**

- The paper addresses the critical and high-impact problem of smart contract security. It maintains a clear focus on a specific, challenging class of bugs: erroneous control flow vulnerabilities (e.g., reentrancy, unsafe external calls, and delegatecall).
- The ablation study (Table 2) is a strong point of the paper. It clearly demonstrates that the proposed method is effective.
- The core idea of combining structural (CFG), syntactic (AST), and human-semantic (Comments) information is logical and provides an intuitive, holistic view for understanding complex code vulnerabilities.

**Weaknesses:**

- The novelty of this paper is limited. The proposed approach is largely an application of existing, standard components (BERT, GCN, GRU, CNN). The fusion mechanism appears to be simple feature concatenation ("vertically stacked"). More discussions are required to highlight its specific novelty over contemporaneous multimodal vulnerability detection work (e.g., Jie et al., 2023; Qian et al., 2023) cited in its own related work section.
- There lacks experimental comparison to other learning-based SOTA methods for vulnerability detection (e.g., Peculiar, or the other GNN/multimodal approaches mentioned in the related work).
- Some experimental results require deeper discussions. E.g., in Table 3, the reported 0% accuracy for both Slither and Mythril on "Access Control" vulnerabilities is puzzling, as these tools are industry standards specifically designed to find such flaws. The authors did not clarify how analysis failures (e.g., contracts that Slither or Mythril failed to parse) were handled in the metrics. Were they excluded, or counted as False Negatives?
- The reported 99.13% accuracy (Table 2) seems high and may indicate overfitting. The paper mentions using SMOTE (Section 4.2) to balance the dataset; it is critical to clarify that SMOTE was applied only to the training split. If synthetic samples from the test set's distribution were included in training (a common data leakage pitfall), the validation and test results would be artificially inflated.

**Questions:**

The paper exhibits several presentation issues that affect clarity and precision. In Section 3.2.1, the authors refer to 256-dimensional vectors from BERT while also describing 128-dimensional node feature vectors in Equation (3), but the relationship between the two is unclear. Moreover, Sections 3.3 and 3.4 reuse the same variable ($F_{ast}$) to represent feature vectors for both the AST and the comments, which may cause confusion.

---

### Official Review · Reviewer_45AM · 2025-10-31

**Soundness:** 3
**Presentation:** 3
**Contribution:** 3
**Rating:** 4
**Confidence:** 1

**Summary:**

The paper introduces a multimodal deep-learning framework to detect erroneous control-flow vulnerabilities in Ethereum smart contracts and to perform contract-level clone detection. It fuses three complementary views into a single contract representation used for verification and similarity search. The authors claim the first application of multimodal deep learning to this class of smart-contract vulnerabilities, outline dataset usage, and note that resources will be open-sourced.

**Strengths:**

- The paper’s multimodal design yields a clearly superior representation, with the full fusion outperforming all single and dual modalities.

- It delivers large, consistent gains over strong baselines across multiple vulnerability types and also shows good transfer to a new dataset, evidencing robustness and generalization.

- Beyond detection, the system adds a practical clone-detection pipeline using the unified contract embedding with an RBF–cosine similarity, broadening utility for auditing and analysis workflows.

**Weaknesses:**

- The ablation exploration is narrow. It focused mainly on learning-rate sweeps and modality combinations, without probing other impactful choices.

- Baseline coverage is thin and clone-detection evaluation hinges on a single dataset and heuristic similarity thresholds.

- The paper provides no theoretical analysis to complement its empirical results.

**Questions:**

- Authors classify a contract as vulnerable when the sigmoid probability exceeds 0.95. Why 0.95? How sensitive are results to that choice?

- Tables report point metrics. Please add variance estimates.

- Could you also probe hidden sizes, dropout, training epochs/early-stopping, and fusion strategies to assess robustness and design choices?

- Please expand to include more learning-based detectors or recent multimodal methods.

---

### Official Review · Reviewer_nHAj · 2025-11-01

**Soundness:** 3
**Presentation:** 2
**Contribution:** 3
**Rating:** 6
**Confidence:** 3

**Summary:**

This paper introduces MultiCFV, a multimodal deep learning framework for detecting control flow vulnerabilities and code clones in smart contracts. The proposed approach integrates three complementary feature types, Control Flow Graphs (CFGs) extracted from bytecode, Abstract Syntax Trees (ASTs) from source code, and comment-based semantic embeddings, to capture structural, syntactic, and contextual information. The authors employ GRU-GCN for graph embedding, CNN with attention for comment feature extraction, and a fusion network for final classification. Extensive experiments are conducted on four benchmark datasets, showing that MultiCFV outperforms existing static analysis tools such as Slither and Mythril in both accuracy and generalization.

**Strengths:**

1. About design. Combining CFG, AST, and comment information is original and addresses the limitations of unimodal vulnerability detectors.
2. About experiments. The work includes comparisons with several baselines, ablation experiments, and cross-dataset evaluation, establishing strong empirical support.
3. High performance. The model achieves good accuracy and generalization to unseen vulnerabilities, including unprotected Ether withdrawal cases.

**Weaknesses:**

1. Incremental novelty. While the multimodal fusion is valuable, it mainly combines known feature extraction techniques rather than introducing a fundamentally new learning paradigm.
2. Limited theoretical justification. The paper lacks a formal explanation of why multimodal integration improves detection robustness beyond empirical evidence.
3. Dataset dependence. The evaluation relies heavily on public datasets; no large-scale or real-world deployment test is included.
4. Scalability and runtime cost. Although mentioned briefly, there is no quantitative analysis of inference time or computational overhead on large-scale contracts.

**Questions:**

1. How does MultiCFV handle unseen vulnerability types not represented in the training set?
2. Could you provide runtime benchmarks or scalability analysis compared with Slither or Mythril?
3. How sensitive is the model to the quality or availability of comments? If comments are sparse or missing, does performance degrade significantly?
4. Were any measures taken to mitigate overfitting given the relatively small vulnerability datasets?
5. Can MultiCFV be adapted for on-chain real-time contract auditing or incremental analysis during contract updates?

---

### Meta-Review · Area_Chair_fmjy · 2026-01-07

**Summary:**

The authors have not prepared a rebuttal. I hope everything is fine with them.

The reviewers see the paper as relevant and on a high-impact problem, but they comment that the contribution is incremental and that several important analyses and comparisons are missing.

**Reviewer Concerns:**

nHAj
- Incremental novelty and missing theoretical explanations.
- No runtime or scalability analysis.
- Generalization to unseen vulnerabilities is unclear.

45AM
- Limited ablation and sensitivity analysis, including the vulnerability threshold.
- Narrow clone detection evaluation and missing variance reporting.
- No theoretical analysis.

Neod
- Insufficient comparison with learning based SOTA.
- Questionable experimental results and potential data leakage.
- Presentation and notation issues.

JHuL
- No comparison with LLM based approaches.
- Missing efficiency measurements.
- Incomplete architectural details and writing issues.

**Reviewer Scores:**

One reviewer rates the paper at 6, three reviewers rate it at 4. With no rebuttal, the scores would not change.

---

### Decision · Program_Chairs · 2026-01-26

Reject